# CAMEx: Curvature-aware Merging of Experts

**Dung V. Nguyen**[1][*]   **Minh H. Nguyen**[1][*]   **Luc Q. Nguyen**[2][*]   **Rachel S.Y. Teo**[3]
**Tan M. Nguyen**[3][†]   **Linh Duy Tran**[2][†]

[1]Faculty of Applied Mathematics and Informatics, Hanoi University of Science and Technology
[2]Viettel AI, Viettel Group
[3]Department of Mathematics, National University of Singapore
`{dung.nv232215M, minh.nh232331M}@sis.hust.edu.vn`
`{lucnq1,linhtd15}@viettel.com.vn`
`rachel.tsy@u.nus.edu,tanmn@nus.edu.sg`

## Abstract

Existing methods for merging experts during model training and fine-tuning predominantly rely on Euclidean geometry, which assumes a flat parameter space. This assumption can limit the model's generalization ability, especially during the pre-training phase, where the parameter manifold might exhibit more complex curvature. Curvature-aware merging methods typically require additional information and computational resources to approximate the Fisher Information Matrix, adding memory overhead. In this paper, we introduce CAMEx (**C**urvature-**A**ware **M**erging of **Ex**perts), a novel expert merging protocol that incorporates natural gradients to account for the non-Euclidean curvature of the parameter manifold. By leveraging natural gradients, CAMEx adapts more effectively to the structure of the parameter space, improving alignment between model updates and the manifold's geometry. This approach enhances both pre-training and fine-tuning, resulting in better optimization trajectories and improved generalization without the substantial memory overhead typically associated with curvature-aware methods. Our contributions are two-fold: (1) CAMEx significantly outperforms traditional Euclidean-based expert merging techniques across various natural language processing tasks, leading to enhanced performance during pre-training and fine-tuning; (2) we introduce a dynamic merging architecture that optimizes resource utilization, achieving high performance while reducing computational costs, facilitating efficient scaling of large language models.

## 1 Introduction

Sparse Mixture of Experts (SMoE) has emerged as a key strategy for constructing large-scale language models (LLMs) by selectively activating model parameters to enhance computational efficiency and scalability. First introduced by Shazeer et al. (2017), SMoE has undergone extensive advancements in both routing mechanisms and expert architecture design. To stabilize token routing, Dai et al. (2022) proposed a two-phase training approach that reduces fluctuations in expert selection during inference. Zhou et al. (2022) introduced expert-choice routing to balance computational loads across experts, while methods to address representation collapse have included cosine scoring (Chi et al., 2022) and fixed random-initialized routers (Chen et al., 2023). Recent innovations have framed routing as optimization or reinforcement learning problems, further refining model efficiency. In terms of expert design, shared expert strategies (Rajbhandari et al., 2022; Dai et al., 2024) allow tokens to engage two experts per layer without increasing communication costs. Additionally, Muqeeth et al. (2024) proposed merging experts by taking weighted averages of their parameters based on router scores, a technique further extended for pretraining and fine-tuning tasks in causal language modeling (He et al., 2023; Zhong et al., 2024; Li et al., 2024). These advancements underscore the versatility and ongoing evolution of SMoE in tackling the computational challenges of scaling LLMs.

---

[*]Equal contribution
[†]Co-corresponding authors

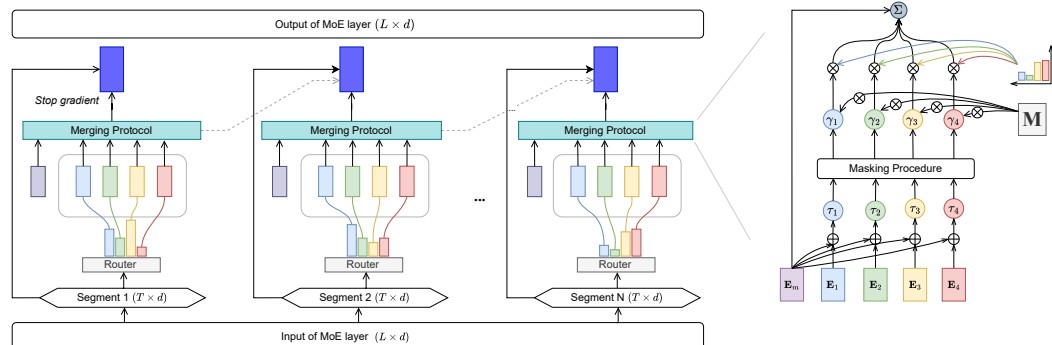

Figure 1: Overview of CAMEx for a causal language modeling SMoE. The experts are merged through the router scores and the curvature-matrix $\mathbf{M}$. During the merging protocol, we can generate the masks for the domain-vectors, denoted as $\gamma_i$, such as Ties or Dare. We follow the causal segmenting pipeline from (Zhong et al., 2024) to achieve both memory efficiency and causal information constraints. Note that *stop gradient* operator is applied for the first segment router scores.

Among existing rigorous research on SMoE, our work focuses on the experts merging lines of research. Specifically, we systemically integrate natural gradient into task-specific merging protocol for SMoE. To the best of our knowledge, the current merging protocol applied for SMoE still deems the parameter space of the expert's parameters as Euclidean ones. Nevertheless, it has been shown that the space of neural network parameters brings the characteristic of the Riemannian manifold (Amari, 1998). Therefore, it is natural for us to make an effort in such a direction for merging experts. Although some existing works on merging models have already leveraged the Fisher Information Matrix (Matena & Raffel, 2022; Jin et al., 2023), we find that they require large computational space and complicated steps to perform well. In contrast, our merging protocol is simple and straightforward to implement while still taking into account the curvature of the parameters manifold. We discover the superior performance of curvature-aware merging in our method compared to the regular merging procedure applied to SMoE. Our main contribution is two-fold:

1. We introduce a novel rapid and efficient merging technique named Curvature- Aware Merging of Experts (CAMEx) for SMoE that includes information about the curvature of the expert's parameters manifold.
2. We propose a new architecture based on CAMEx, which dynamicalizes the merging protocol along with parameters reduction. Our empirical experiments prove the dominant performance of this architecture on pre-training tasks.

We empirically demonstrate that 1) our proposed merging method can add in rapidness of convergence speed for pre-training and 2) when combined with other merging protocols, it can boost the model's performance on a variety of practical tasks, including language modeling, text classification, question answering, and image classification.

## 2 CURVATURE-AWARE MERGING OF EXPERTS

### 2.1 BACKGROUND: EXPERT MERGING IN SPARSE MIXTURE OF EXPERTS

It is convenient to recall the concept of SMoE and a few well-known experts merging methods for SMoE. From this point to the rest of our paper, let us use the notations summarized in Table 1.

**Sparse Mixture of Experts.** A SMoE layer processes the tokens series as follows:

$$\begin{cases} \mathbf{y}_t = \sum_{i \in \mathcal{S}_t} \mathbf{G}(t, i) \cdot \mathbf{E}_i \mathbf{h}_t \\ \mathbf{G}(t, \cdot) = \text{softmax}(\mathbf{W}_g \mathbf{h}_t) \\ \mathcal{S}_t = \text{top-k}(\mathbf{G}(t, \cdot)) \end{cases} \tag{1}$$

**SMEAR.** Muqeeth et al. (2024) introduces the ensemble of expert parameters through weighted average computation with the factors are the router scores.

Table 1: Notations and Definitions.

| Symbol | Description | Symbol | Description |
|---|---|---|---|
| $T$ | Number of tokens | $k$ | Number of selected experts |
| $N$ | Total number of experts | $\mathbf{E}_i \in \mathbb{R}^{d \times h}$ | Weights for the $i$th expert |
| $\mathbf{h} \in \mathbb{R}^{T \times d}$ | Input tokens or hidden states | $\mathbf{G}(\cdot, \cdot) \in \mathbb{R}^{T \times N}$ | Gating function output |
| $\mathcal{S}_t$ | Set of top-k experts for token $\mathbf{h}_t$ | $\alpha \in [0, 1]$ | Rescalling factor |

**Task-Specific merging in SMoE.** Our work will follow the scheme of task-specific merging (Ilharco et al., 2023). In such a setting, we assume the existence of $N$ pre-trained models parameterized by $\theta_i$ each was pre-trained on a different task. We then define the task-vector for each pre-trained model through the merged model $\theta_m$ as $\tau_i = \theta_i - \theta_m$. The merging protocol will be performed by Eqn. Merg. Under the context of SMoE, each expert learns to handle a particular subset of the input space or specializes in a specific type of feature or pattern (Jacobs et al., 1991; Dai et al., 2024). We believe it is more suitable to reference this technique as **domain-specific merging**. We, therefore, will rename the tensors $\tau_i = \mathbf{E}_i - \mathbf{E}_m$ as *domain-vector*. Additionally, to take the router information into account, we will define the formulation for domain-specific merging in a SMoE layer as follows:

$$\hat{\mathbf{E}}_m = \mathbf{E}_m + \alpha \sum_{i=1}^{N-1} s_i \tau_i \tag{2}$$

where $s_i$ denotes the score of the router for the $i$th expert. We want to note that with $0 < \alpha < 1$, domain-specific merging aligns with soft merging.

## 2.2 GRADIENT INTERPRETATION OF MODELS MERGING

We want to emphasize the alignments between the paradigm of gradient descent and model merging. For this, we denote $\theta \in \mathbb{R}^N$, $\mathcal{L}(\theta)$, and $\eta$ as the model's parameters, the empirical loss function, and the learning rate, respectively. During the training process of a deep learning model, the parameters are updated following the gradient descent formula:

$$\theta_{n+1} = \theta_n + \eta(-\nabla \mathcal{L}(\theta_n)) \tag{GD}$$

In the aspect of deep models merging, we also have an update rule in a similar manner, which is

$$\hat{\theta}^m = \theta^m + \alpha \sum_{i=1}^{n} \underbrace{(\theta^i - \theta^m)}_{\substack{\text{gradient-like} \\ \text{update direction}}} \tag{Merg}$$

where $\theta^m$ denotes the merged model's parameters, and $\theta^i$ denotes the parameters of the $i$th expert. Here, we interpret $\theta^i$ as the optimal parameters of the model for a specific task or domain, and then the update rule gives us a direction toward optimizing for all tasks.

However, it has been pointed out by Amari (1998) that the parameter space structure of deep learning models has Riemannian characteristics. Therefore, a more *natural* gradient updating scheme was proposed,

$$\theta_{n+1} = \theta_n + \eta \underbrace{G(\theta_n)(-\nabla \mathcal{L}(\theta_n))}_{\text{natural gradient}} \tag{NGD}$$

In this formula, $G(\theta_n) \in \mathbb{R}^{N \times N}$ denotes the *Riemannian metric tensor* (Amari, 1998; Amari & Douglas, 1998), which characterizes the intrinsic curvature of a particular manifold in $N$-dimensional space (Martens, 2020) or sometimes, the inversed Fisher Information Matrix. The same ideology was introduced for merging large language models in Fisher merging (Matena & Raffel, 2022) and Regmean (Jin et al., 2023). However, both methods suffer from the bottleneck in the computation cost of approximating the Fisher Information.

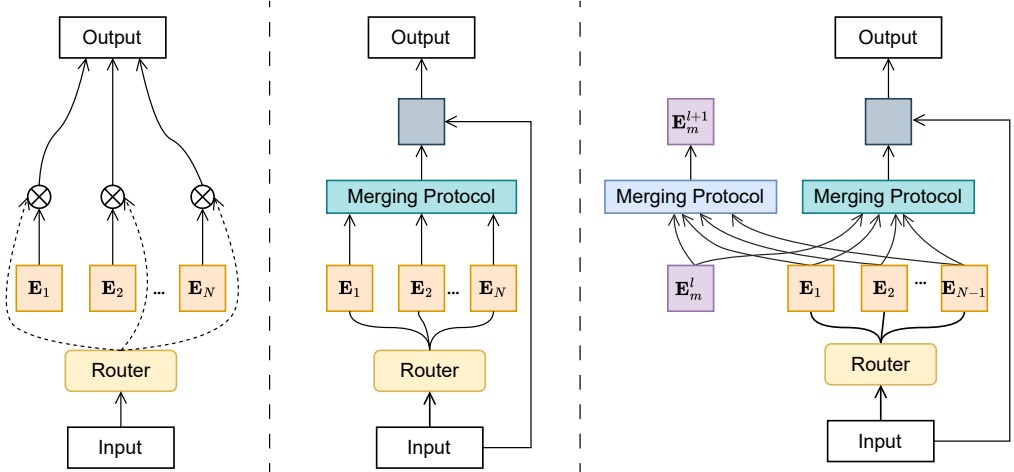

Figure 2: **Overall architecture of different SMoE layers**. The figure presents the vanilla SMoE layer on the left, the merging expert layer in the middle, and our proposed dynamic merging SMoE layer on the right. Our architecture reduces the number of parameters compared to the other two, while maintaining the same number of activated neurons per layer. Importantly, despite the dynamic merging mechanism, our architecture preserves the same number of experts at each layer as the other SMoE architectures, ensuring comparable model capacity, i.e., the number of activated parameters per layer.

## 2.3 EXPERTS MERGING WITH CURVATURE-AWARE

We introduce an efficient way to merge experts within SMoE layers, based on the causal segmenting approach proposed by (Zhong et al., 2024). The goal of the causal segment routing strategy is to enhance the efficiency of expert merging operations while maintaining the autoregressive nature of language models. More details about this algorithm can be found in Appendix B.1 and Algorithm 1. We then perform the following merging protocols:

$$\hat{\mathbf{E}}_m^l = \mathbf{E}_m^l + \alpha \sum_{i=1}^{N-1} \mathbf{M}_i \cdot (s_i^l * \tau_i^l) \tag{CA-Merg}$$

where $\mathbf{M}_I \in \mathbb{R}^{d_{in}d_{out} \times d_{in}d_{out}}$ denote the curvature matrix which performs matrix product with the gradient-like component. The curvature of the parameters manifold will be learned through these tensors while optimizing the empirical loss. This approach has also proven its effectiveness in meta-learning for few-shot classification (Park & Oliva, 2019). We further explore the computing efficiency of merging experts by proposing a novel dynamic merging formula

$$\begin{cases} \mathbf{E}_m^{l+1} &= \mathbf{E}_m^l + \dfrac{\alpha}{N-1} \sum_{i=1}^{N-1} \mathbf{M}_i \cdot \tau_i^l \\ \hat{\mathbf{E}}_m^{l+1} &= \mathbf{E}_m^{l+1} + \alpha \sum_{i=1}^{N-1} \mathbf{M}_i \cdot (s_i^{l+1} * \tau_i^{l+1}) \end{cases} \tag{Dynamic-Merg}$$

The architecture corresponding to this recurrent representation can be found in Figure 2. The architecture contains a global expert that traverses through the SMoE layers by the updating rule in Eqn. Dynamic-Merg. Not only will this allow a notable reduction in model size and GFLOPS, but it also ensures the number of experts in each SMoE is the same as in the full-expert setting, where each layer has the same number of experts.

## 2.4 EFFICIENCY

**Parameter efficient approximation of curvature matrix.** Storing and computing a curvature matrix requires a whopping memory and time complexity of $O(n^4)$ and $O(n^4)$, respectively. This is infeasible even for a simple SMoE layer, as one layer can contain many experts. To mitigate

Table 2: **Performance of T5-base variants on the fine-tuning tasks for GLUE**. All SMoE variants have 8 experts per layer. We follow Devlin et al. (2019) in conducting experiments on the GLUE benchmark. Our curvature-aware methods outperform all baselines across tasks, while maintaining the same number of parameters and FLOPs as the SMoE models.

| Methods | Params | TFLOPs | SST-2 | MRPC | CoLA | QQP | STSB | QNLI | RTE | MNLI |
|---|---|---|---|---|---|---|---|---|---|---|
| Vanilla | 220M | 4.65 | 93.34 | 89.70 | 58.06 | 88.76 | 89.06 | 92.34 | 74.36 | 86.36 |
| SMoE | 1.0B | 4.65 | 94.26 | 90.87 | 56.78 | 88.69 | 89.44 | 92.07 | 70.75 | 86.38 |
| Domain-Specific | 1.0B | 4.65 | 93.57 | 90.19 | 58.07 | 88.77 | 89.40 | 92.51 | 72.56 | 86.40 |
| Ties | 1.0B | 4.65 | 93.92 | 91.44 | 58.54 | 86.47 | 88.58 | 91.87 | 75.54 | 86.39 |
| Dare | 1.0B | 4.65 | 93.80 | 89.46 | 58.33 | 88.72 | 89.13 | 92.29 | 73.64 | 86.20 |
| **Domain-specific-CA** | 1.0B | 4.65 | 93.80 | 91.16 | 58.57 | **88.86** | 89.47 | 92.60 | 74.72 | 86.44 |
| **Dare-CA** | 1.0B | 4.65 | 94.49 | 91.15 | 58.56 | 88.76 | **89.56** | **92.80** | **78.70** | 86.34 |
| **Ties-CA** | 1.0B | 4.65 | **94.61** | **92.49** | **60.06** | 88.83 | 89.54 | 91.89 | 75.81 | **86.45** |

this problem, we follow Martens & Grosse (2015) and approximate the curvature matrix using the Kronecker product. It has been proven by Hameed et al. (2022) that we can approximate arbitrary matrix using a finite sum of Kronecker products. For a curvature matrix $\mathbf{M}_i \in \mathbb{R}^{d_{in}d_{out} \times d_{in}d_{out}}$, we present the rank-1 approximation as below:

$$\mathbf{M}_i \approx \mathbf{M}_i^{in} \otimes \mathbf{M}_i^{out} \tag{3}$$

with $\mathbf{M}_i^{in} \in \mathbb{R}^{d_{in} \times d_{in}}$ and $\mathbf{M}_i^{out} \in \mathbb{R}^{d_{out} \times d_{out}}$. Still, this form of approximation is too large to compute and store during training time, so we further decompose $\mathbf{M}_i^{in}$ and $\mathbf{M}_i^{out}$ using Kronecker product because of the efficient computation using tensor algebra. This form of approximation reduces the number of parameters added and only puts negligible memory and computational overhead to the training process at the cost of additional $O(n)$ memory complexity and $O(n^{2.5})$ computational complexity. Although this might limit the representative capacity of the curvature matrix, we empirically find that the performance of our method still surpasses other merging methods.

**Efficient test-time inference with reparameterization.** We focus on the case where $\alpha = 1$. To further optimize the computation of curvature-aware merging, we embed the curvature matrices into the domain-vectors using the following reparameterization trick:

$$\mathbf{E}_i' \leftarrow \mathbf{E}_m + \mathbf{M}_i \cdot \tau_i \tag{4}$$

In this case, the merging formula at test time becomes:

$$\hat{\mathbf{E}}_m = \mathbf{E}_m + \sum_{i=1}^{N-1} s_i \cdot (\mathbf{E}_i' - \mathbf{E}_m) = \mathbf{E}_m + \sum_{i=1}^{N-1} \mathbf{M}_i \cdot (s_i \cdot \tau_i)$$

Thus, during inference, we avoid storing the curvature matrices and recomputing their product with domain vectors, reducing the total FLOPs. This explains the computational efficiency seen in Section 3.

## 3 EXPERIMENTAL RESULTS

In the following tables, the row with our method's results is highlighted in *grey*. Results with the best and second best performance are written in **bold** and underline, respectively. In addition, methods with the postfix "-CA" denote the curvature-aware version of the corresponding baseline. In Table 2, CA-augmented models consistently outperform non-CA versions, with Ties-CA achieving the highest scores on SST-2 (94.61), MRPC (92.49), CoLA (60.06), and MNLI (86.45). Similarly, Dare-CA outperforms Dare on RTE

Table 3: Performance of GPT-2 small variants on Wikitext-2 and wikitext-103

| Methods | Perplexity↓ | Params (M) | GFLOPS↓ |
|---|---|---|---|
| **Wikitext-103 Results** | | | |
| Vanilla | 23.03 | 125 | 292.5 |
| SMoE | 22.42 | 522 | 292.5 |
| Domain-specific | 21.64 | 522 | 292.5 |
| **Domain-specific-CA** | **21.50** | 522 | 292.5 |
| **Dynamic** | 21.55 | **470** | 292.5 |
| **Wikitext-2 Results** | | | |
| Vanilla | 21.84 | 125 | 292.5 |
| SMoE | 21.60 | 522 | 292.5 |
| Domain-specific | 21.56 | 522 | 292.5 |
| Ties | 21.45 | 522 | 292.5 |
| Dare | 21.60 | 522 | 292.5 |
| **Domain-specific-CA** | **21.06** | 522 | 292.5 |
| **Dare-CA** | 21.42 | 522 | 292.5 |
| **Ties-CA** | 21.11 | 522 | 292.5 |

(78.70), showing CA's benefit on smaller, more variable datasets, while Domain-specific-CA surpasses its non-CA counterpart on QNLI and MNLI. Table 3 shows that Domain-specific-CA achieves the lowest perplexity (21.50 on Wikitext-103, 21.06 on Wikitext-2), outperforming all other methods. The Dynamic model follows closely with a perplexity of 21.55 while using 9% fewer parameters, making it more efficient. Notably, curvature-aware (CA) methods consistently enhance performance without increasing computational cost. For example, Ties-CA (21.11) and Dare-CA (21.42) significantly improve over their respective baselines. In contrast, while SMoE and other non-CA methods provide some gains over the Vanilla model, their improvements are relatively minor despite the increased parameter count.

## 4 RELATED WORK

**Sparse Mixture-of-Experts (SMoE).** As the demand for model scaling grows increasingly widespread, there is a pressing inquiry into efficient ways to optimize computing costs while minimizing the impact on model performance. To address this need, Sparse Mixture of Experts (SMoE) has emerged and undergone extensive research and exploration (Shazeer et al., 2017; Lepikhin et al., 2021; Fedus et al., 2022). Starting with Shazeer et al. (2017), the integration of SMoE into transformer architectures followed shortly after with the works of Lepikhin et al. (2021) and Fedus et al. (2022). The principle of SMoE is based on a simple concept: scaling the horizontal dimension of models (i.e., the number of feedforward blocks) rather than the vertical dimension (i.e., the number of stacked layers). This allows the model to selectively activate units or parameters based on the input tokens, thereby optimizing resource usage while maintaining performance.

**SMoE Efficiency Bottlenecks and Emerging Solutions.** While it remains controversial whether to use Top-1 or Top-K routing, some research has highlighted the potential performance gains from increasing the number of activated experts (Shazeer et al., 2017; Chen et al., 2023). Other studies have found redundancies among experts in MoE layers (Li et al., 2024; Lu et al., 2024a). Additionally, some work has proposed using low-rank experts (Wu et al., 2024b; Liu et al., 2024; Wu et al., 2024a) inspired by LoRA (Hu et al., 2022). Despite the varying research directions, these studies consistently show that training a robust SMoE requires substantial computational and memory resources. This has motivated researchers such as Li et al. (2024), He et al. (2023), and Zhong et al. (2024) to merge experts within each MoE layer, reducing the number of experts to a single one and significantly improving training and inference efficiency.

**Model Merging with curvature-aware.** Though numerous methods for merging models have been introduced and developed (Yadav et al., 2023; Cai et al., 2023; Ilharco et al., 2022; Matena & Raffel, 2022; Jin et al., 2022; Don-Yehiya et al., 2022; Rame et al., 2023; Lu et al., 2024b), most of these works consider merging protocols in the Euclidean parameter space. However, it has been noted that the space of deep neural network models is a Riemannian one (Amari, 1998). Matena & Raffel (2022) and Jin et al. (2022) were the first to fuse model weights while accounting for the Fisher Information. Despite their promising results, these methods require massive computation to approximate the inversion of the Fisher matrix. Moreover, the Fisher matrix has a size proportional to the dimension of the model parameters, which significantly increases memory usage. Consequently, these methods are challenging for directly integrating into SMoE layers to fuse expert weights.

## 5 LIMITATION AND CONCLUSION

We introduced CAMEx, a curvature-aware approach to expert merging in Mixture of Experts architectures. By leveraging natural gradients to account for parameter manifold curvature, CAMEx improves performance and reduces computational costs during pre-training and fine-tuning, outperforming traditional Euclidean-based methods. Our dynamic architecture incorporates a global expert across layers, optimizing resource usage and minimizing model size without compromising accuracy. A minor limitation is reduced compatibility with Ties and Dare merging at higher Kronecker ranks, which future research could address while exploring broader curvature-aware applications in neural networks. This work advances efficient and scalable model development for large-scale machine learning.

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

# Supplement to "CAMEx: Curvature-aware Merging of Experts"

## Table of Contents

## A ADDITIONAL DETAILS ON DATASETS

This section provides detailed information on the datasets and evaluation metrics used in the experiments in Section 3.

### A.1 LANGUAGE MODELING ON WIKITEXT

**The WikiText-103** dataset consists of Wikipedia articles designed to capture long-range contextual dependencies. The training set includes approximately 28,000 articles, totaling around 103 million words. The validation and test sets have 218,000 and 246,000 words, respectively, spread across 60 articles per set, with each set comprising roughly 268,000 words. Our experiments follow the standard procedure described in Merity et al. (2017).

**WikiText-2** is a smaller version of WikiText-103, containing 2 million tokens and a vocabulary of 33,000 words.

### A.2 TEXT CLASSIFICATION ON GLUE BENCHMARK

These tasks include MNLI (Williams et al., 2018), which assesses a model's ability to determine entailment between pairs of sentences; QQP (Quora, 2017) and MRPC (Dolan & Brockett, 2005),

which focus on identifying sentence similarity and paraphrase detection; SST-2 (Socher et al., 2013) for sentiment analysis; CoLA (Warstadt et al., 2019) for grammaticality judgment; and QNLI (Wang et al., 2019) for question-answer classification. Additionally, STSB (Cer et al., 2017) evaluates the model's ability to measure sentence similarity, while RTE (Dagan et al., 2006) tests logical reasoning.

## A.3 Question-answering on SQuAD and WikiQA

**SQuADv1.1** (Rajpurkar et al., 2016) (Stanford Question Answering Dataset) is a widely used benchmark for reading comprehension and question answering tasks. It contains over **100,000 question-answer pairs** sourced from more than 500 Wikipedia articles. Each question is paired with a paragraph from the article, where the answer is a span of text extracted from the passage. The dataset consists of natural language questions that cover a wide range of topics, context paragraphs from Wikipedia, and answers marked by their start and end positions within the context. The primary task is to extract the correct answer span based on the posed question. Key features of the dataset include the need for exact span extraction, the large dataset size, and its task design focused on reading comprehension. Evaluation is typically done using Exact Match (EM), which measures the percentage of predictions that exactly match the ground-truth answers, and the F1 score, which measures the overlap between predicted and ground-truth answers by calculating the harmonic mean of precision and recall.

**WikiQA** (Yang et al., 2015) is an open-domain question answering dataset designed for answer sentence selection tasks. It consists of natural language questions primarily extracted from search engine queries, with candidate sentences sourced from Wikipedia articles. Each candidate sentence is labeled as either a correct or incorrect answer for the given question. The dataset contains 3,047 questions and 29,258 candidate sentences. The main challenge is selecting the correct sentence from a set of candidates, unlike SQuADv1.1, where the task focuses on extracting a text span. Key features include its real-world query origins, the sentence selection task, and the open-domain nature, which requires models to identify relevant sentences from diverse topics. WikiQA is evaluated using Accuracy.

## A.4 Image Classification on Imagenet

**ImageNet-1k**, the most widely utilized subset of the ImageNet dataset introduced by Deng et al. (2009), comprises 1.28 million images for training and 50,000 images for validation, across 1,000 categories. Performance evaluation is typically based on top-1 and top-5 accuracy metrics.

## A.5 Adversarial Examples and Out-of-distribution datasets

**ImageNet-A**: The ImageNet-A dataset (Hendrycks et al., 2021b) contains real-world images specifically curated to fool ImageNet classifiers. It focuses on 200 classes, a subset of the 1,000 classes in ImageNet-1k. Errors made within these 200 classes are considered particularly significant, as they represent a wide variety of categories from ImageNet-1k.

**ImageNet-O**: This dataset consists of examples adversarially filtered to challenge out-of-distribution (OOD) detectors on ImageNet (Hendrycks et al., 2021b). It includes images from the larger ImageNet-22k dataset but excludes those present in ImageNet-1k. The selected samples are those that a ResNet-50 model confidently misclassifies as an ImageNet-1k class, and the primary evaluation metric is the area under the precision-recall curve (AUPR).

**ImageNet-R**: ImageNet-R contains a variety of artistic renditions of object classes found in the original ImageNet dataset (Hendrycks et al., 2021a). This dataset includes 30,000 artistic representations of images from 200 classes, selected from the ImageNet-1k subset. The dataset was created to challenge models with non-standard visual interpretations of the classes.

---

**Algorithm 1** The Overall Procedures of CAMEx.

---

1: **Initialize:** A model $\mathcal{M}$ with $l$ SMoE layers, the total number of original experts $N$.
2: Let $\mathtt{H}^t \in \mathbb{R}^{B \times L \times N}$ and $\mathtt{T}^t \in \mathbb{R}^{B \times L \times d}$ denote the *router logits* and the *sequence of tokens* at intermediate layer $t$, respectively.
3: **for** layer $t = 1, \ldots, l$ **do**
4:     $K = L/S, T^l \leftarrow \text{RESHAPE}(T, B * K, S, d)$             ▷ Begin Causal Segmenting
5:     $\mathtt{H}^l \leftarrow \mathbf{G}\left(T^l\right)$
6:     $\mathtt{H}^l \leftarrow \text{ROLLandDETACH}\left(\mathtt{H}^l\right)$
7:     **if** TIES-MERGING **then**             ▷ Generate mask for merging
8:         **for** expert $i = 1, \ldots, N - 1$ **do**
9:             $\tau_i \leftarrow \mathbf{E}_i - \mathbf{E}_m$
10:             $\gamma_i \leftarrow sgn(\tau_i)$
11:         **end for**
12:         $\gamma^m = sgn(\sum_{i=1}^{N-1} \tau_i)$
13:         **for** expert $i = 1, \ldots, N - 1$ **do**
14:             $\tau_i^m \leftarrow \gamma_i \wedge \gamma^m$
15:             $\tau_i \leftarrow \tau_i \cdot \mathbf{M}_i$
16:         **end for**
17:     **else**
18:         Generate mask for DARE-MERGING
19:     **end if**
20:     $\mathbf{E}_m \leftarrow \mathbf{E}_m + \gamma_m * \sum_{i=1}^{N-1} \mathtt{H}_i^l * \tau_i$             ▷ Merge Experts
21: **end for**

---

## B    Algorithm and implementation details

### B.1    Causal segmenting

Background of Causal Segmenting: A significant advancement in SMoE design centers on fully differentiable architectures that eliminate the need for additional loss terms to stabilize training. In Muqeeth et al. (2024), a model was introduced that computes a weighted average of expert feed-forward networks (FFNs). For an input $x$ with corresponding routing weights, the output is defined as:

$$o_x = \text{FFN}\left(h_x; \sum_{i=1}^{N} s_i \cdot \mathbf{E}_i\right), \quad \text{where} \quad s_i = \text{Softmax}(\mathbf{G}(h_x))_i.$$

However, applying this approach to autoregressive language models is computationally costly, as the merged FFN must be computed for each token in the sequence, leading to costs that scale linearly with the number of experts. An alternative based on pooling—routing via the sequence's average representation, as follows:

$$s_i = \text{Softmax}\left(\mathbf{G}\left(\frac{\sum_{j=1}^{L} h_{x_j}}{L}\right)\right)_i.$$

This, however, disrupts the autoregressive property essential for pre-training. To address this, Zhong et al. (2024) introduced causal segment routing. This technique merges FFNs in an MoE layer by utilizing information from the preceding segment to process the current segment. Specifically, given a training instance $X$ consisting of $L$ tokens (e.g., $L = 4096$), we divide the instance into $N$ segments, each containing $T$ (e.g., $T = 256$) consecutive tokens. For the $k$-th segment $S_k$, where $k > 1$, we compute the average of the hidden representations from the previous segment $S_{k-1}$, denoted as $\bar{h}_{k-1}$. By using the average hidden representation, the model can adapt to prompts of varying lengths during inference. The average hidden representation $\bar{h}_{k-1}$ is then employed to determine the routing weights, leading to a merged expert $\bar{\mathbf{E}}$:

$$\bar{h}_{k-1} = \frac{1}{T} \sum_{x \in S_{k-1}} h_x, \quad s_i = \text{Softmax}(\mathbf{G}(\bar{h}_{k-1})), \quad \bar{\mathbf{E}} = \sum_i s_i \cdot \mathbf{E_i}. \quad (5)$$

The merged expert $\bar{\mathbf{E}}$ is then used to process all tokens in the current segment $S_k$, i.e., $o_x = \text{FFN}(h_x; \bar{\mathbf{E}}), \forall x \in S_k$. This approach ensures that the model's routing decisions rely exclusively on

data from preceding positions. For the first segment $S_1$, the segment's own representation is used to compute the merging weights for its FFN. To prevent information leakage, a stop-gradient operation is applied to $\mathbf{G}(\bar{h}_1)$:

$$\bar{h}_0 = \frac{1}{T} \sum_{x \in S_0}^{T} h_x \tag{6}$$

These tokens are then used to calculate the scores for the merging procedure

$$
\begin{aligned}
s_0 &= \text{DETACH}\big(\mathbf{G}(\bar{h}_1, k)\big) \\
s_i &= \mathbf{G}(\bar{h}_{i-1}), \quad i = 1, \ldots, S-1
\end{aligned}
\qquad \text{(ROLLandDETACH)}
$$

### B.2 SOME IMPLEMENTATIONS

**Implementation of Kronecker product** We consider the case where experts are linear layers

```
# Calculating domain-specific vectors
taus = weights - weight_m

# output_size = dim_out1 * dim_out2, input_size = dim_in1 * dim_in2
taus = taus.view(1, -1, dim_out1, dim_out2, dim_in1,
    dim_in2).repeat(rank, 1, 1, 1, 1, 1)
# Calculate Kronecker-product
taus = torch.einsum("rbij, rbjklm->rbiklm", curve1_out, taus)
taus = torch.einsum("rbik, rbjklm->rbjilm", curve2_out, taus)
taus = torch.einsum("rbil, rbjklm->rbjkim", curve1_in, taus)
taus = torch.einsum("rbim, rbjklm->rbjkli", curve2_in, taus)
# Summation along the Kronecker rank dimension and reshape
taus = taus.sum(0)
taus = taus.reshape(-1, output_size, input_size)
```

## C   MORE EXPERIMENT DETAILS

We perform evaluations on **four** major tasks, including language modeling, text classification, question answering, and image classification. For language modeling, we use the Wikitext-2 and Wikitext-103 (Merity et al., 2016) benchmarks. For text classification, we employ a subset of the GLUE (Wang et al., 2019) benchmark, a collection of **eight** diverse tasks designed to test different aspects of language understanding. For question answering, we employ two famous benchmarks: SQuAD (Rajpurkar et al., 2016) and WikiQA (Yang et al., 2015). Finally, the ImageNet-1k (Deng et al., 2009) dataset is chosen for image classification evaluation.

We choose GPT-2 (Radford et al., 2019) small and Swin-Transformer small (Liu et al., 2021) as our backbones for language modeling and image classification, respectively. Regarding GLUE and question-answering tasks, T5 base (Raffel et al., 2020) is chosen.

Our experimental results confirm that the proposed merging method accelerates pre-training convergence and, when combined with other merging protocols, enhances model performance across tasks and settings. All results are averaged over 5 runs with different random seeds. Detailed information on the datasets, models, training procedures, and hyperparameters is provided in Appendix A and Appendix C. For additional experiments on different routers and merging methods, we refer to Appendix D.2, and D.3.

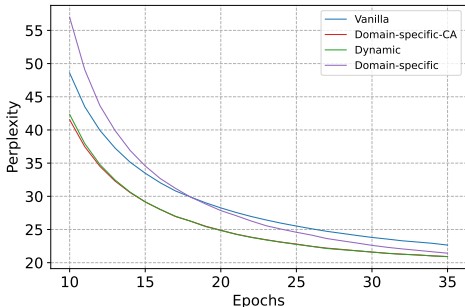

Figure 3: Perplexity of GPT2-small variants starting at the tenth epoch.

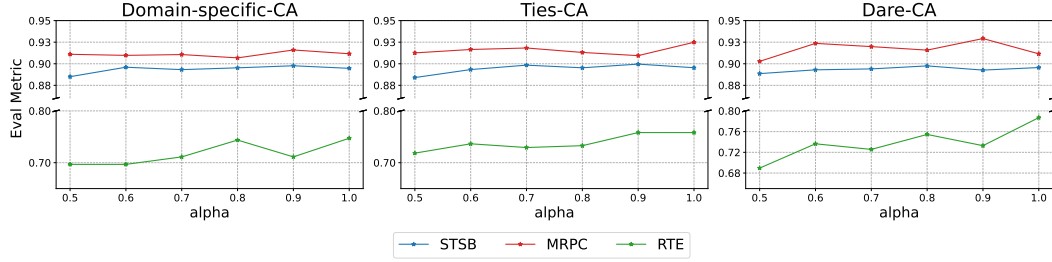

Figure 4: Impact of the $\alpha$ parameter on Curvature-Aware method performance across NLP tasks. We observe that the scaling factors that are within the range $[0.8, 1]$ consistently improve model's performance.

### C.1 TRAINING AND EVALUATION DETAILS

We fix the number of epochs for all models on each task. For each text-related task, we first undertake a comprehensive hyper-parameter search. This encompasses batch sizes from $\{8, 16, 32, 64\}$, learning rates from $\{3e{-}4, 1e{-}4, 3e{-}5, 1e{-}5\}$, to pinpoint the optimal fine-tuned models. Regarding image classification tasks, a batchsize of 96 for chosen for all models. In addition, we choose AdamW (Loshchilov & Hutter, 2019) as the default optimizer and conduct all experiments on NVIDIA A100 GPUs. We compare our proposal to three merging baselines, including domain-specific, Ties, and Dare merging. There exists prior works on merging methods with the aid of the Fisher Information Matrix, such as Matena & Raffel (2022), which rely on access to a validation set used to compute the Fisher matrix or fine-tune hyperparameters. To eliminate the need for a validation set, Jin et al. (2023) proposes storing and transmitting inner product matrices derived from the training data for each task, which are of the same size as the original model. However, this approach becomes costly for large models, as storage and transmission demands increase linearly with model size and the number of tasks as well as the number of experts. Therefore, we choose baselines that are needless of extra information and computational cost to perform comparisons. We want to note that our merging protocol can be easily integrated into other works such as merge then compress protocol Li et al. (2024).

**Supervised Fine-Tuning Hyper-Parameters**  Besides {batch size, learning rate, epoch counts} which vary for each task, we keep other hyper-parameters of supervised fine-tuning fixed for all tasks. These are shown in Table 4.

Table 4: Fine-tuning hyper-parameters of all models in Section 3

| Hyper-Parameters | Values |
| --- | --- |
| Optimizer | ADAMW |
| Adam $\epsilon$ | $1e{-}6$ |
| Adam $\beta$ | $(0.9, 0.98)$ |
| Warm-up steps | 16 |
| Weight decay | 0.01 |
| LR scheduler | LINEAR DECAY |
| Scaling factor $\alpha$ | 1 |
| Kronecker rank $r$ | 1 |

## D ADDITIONAL EXPERIMENTS

### D.1 ABLATION

**Impact of the scaling factor.**  The plot in Figure 4 illustrates the impact of the $\alpha$ parameter on the performance of three curvature-aware (CA) model variants Domain-specific-CA, Ties-CA, and Dare-CA across three natural language processing tasks: STSB, MRPC, and RTE. The $\alpha$ parameter

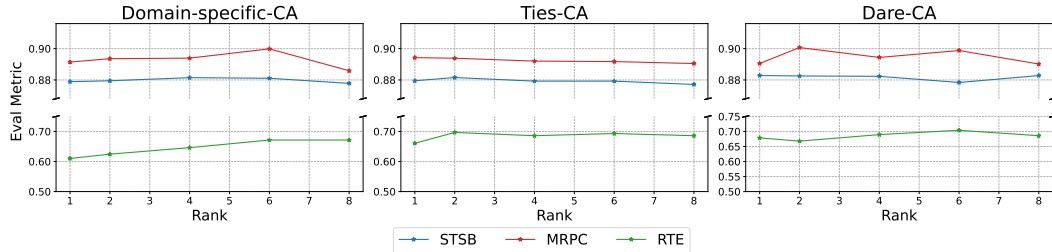

Figure 5: Impact of the Kronecker rank of curvature matrix on model's performance. We observe that as the rank increases the performance drops and then saturates. However, we would like to note that this curve might change depending on the downstream tasks and the merging protocol.

Table 5: Comparison of Accuracy for Swin-Transformer small variants on ImageNet-1k.

| Methods | Params (M) | GFLOPs | Acc@1 | Acc@5 |
|---|---|---|---|---|
| Vanilla | 50 | 6.75 | 83.14 | 96.90 |
| SMoE | 157 | 6.75 | 83.15 | 96.71 |
| Domain-specific | 157 | 6.75 | 83.15 | 96.91 |
| Ties | 157 | 6.75 | 83.28 | 96.93 |
| Dare | 157 | 6.75 | 83.13 | 96.88 |
| **Domain-specific-CA** | 157 | 6.75 | 83.29 | 96.95 |
| **Ties-CA** | 157 | 6.75 | **83.38** | **96.96** |
| **Dare-CA** | 157 | 6.75 | **83.38** | 96.94 |

ranges from 0.5 to 1.0. The overall trend suggests that increasing $\alpha$ leads to better generalization, particularly for complex tasks such as RTE, where sentence-level entailment and similarity benefit from stronger curvature-aware representations. Moreover, across all tasks, the model reaches its peak performance when $\alpha$ is inside the range $[0.8, 1]$. This observation aligns with that indicated by Yadav et al. (2023). For a more comprehensive analysis on the impact of $\alpha$ and number of experts, we direct the readers to Appendix D.5.1 and D.5.2.

**Improved performance with higher Kronecker rank.** Across all three tasks (STSB, MRPC, and RTE), the evaluation metrics tend to improve as the rank increases from 1 to 8. This indicates that higher-ranked models generally perform better, suggesting a positive correlation between rank and task performance. Notably, the Domain-specific-CA model consistently achieves high performance across all tasks, especially in STSB, where metrics approach 0.90. Although MRPC and RTE show slightly lower metrics, ranging from 0.50 to 0.75, there is a clear improvement in performance as rank increases, particularly in the lower-to-mid ranks. However, we observed a decline in performance for Ties-CA and Dare-CA as the rank increases. We hypothesize that this is due to the masking mechanism employed by these methods, which may interfere with the learning process of the curvature matrices.

**Robustness against noise.** Table 6 demonstrates that curvature-aware models offer superior performance on corrupted ImageNet datasets compared to both Vanilla and SMoE variants. Among the models, our best configuration (Ties-CA) stands out as the best performer, showcasing robustness

Table 6: Comparison for Swin-Transformer small variants on corrupted ImageNet.

| Methods | ImageNet-O | ImageNet-A | ImageNet-R |
|---|---|---|---|
| Vanilla | 45.88 | 23.68/53.10 | 37.34/52.34 |
| SMoE | 43.34 | 23.72/53.15 | 38.02/55.17 |
| **Ours** | **50.69** | **25.45/54.24** | **38.37/55.42** |

to corruptions across all datasets. These results suggest that incorporating curvature-awareness can substantially improve model robustness in challenging conditions.

## D.2 INTEGRATING CAMEx INTO TWIN-MERGING

We expand our experiments to include a broader range of most recent merging expert methods. Specifically, we integrated our CAMEx method with the Twin-Merging approach (Lu et al., 2024b). Key distinctions between CAMEx and Twin-Merging lie in their core mechanisms:

- Our method is a non-Euclidean merging method, which utilizes the curvature-aware matrix, whereas Twin-Merging is a model merging method, which relies on Euclidean merging.

- Our approach is specifically designed for finetuning, in contrast to Twin-Merging, which is intended for post-training.

- Finally, our dynamic mechanism performs inter-layer to form the merged expert, unlike Twin-Merging, which uses within-layer pre-calculations for merging. To integrate our method with Twin-Merging, we first fine-tune the Curvature Aware model for a specific GLUE task. At test time, we apply the Twin-Merging algorithm to merge experts, referring to our approach as Twin-CA. Notably, we found Twin-Merging to be a simple yet powerful technique that is easy to implement and helps reduce memory usage during inference. We adhere to the original implementation settings, using a sparsity density value of 0.2.

Table 7: Performance of Twin-Merging and its Curvature Aware (CA) variant on GLUE tasks.

| Method | MRPC | RTE | STSB |
|---|---|---|---|
| Twin-Merging | 91.97 | 72.20 | 88.56 |
| Twin-CA (**Ours**) | **92.30** | **74.73** | **89.55** |

The results in Table 7 demonstrate the effectiveness of our CAMEx approach when integrated with the Twin-Merging mechanism on GLUE tasks, highlighting its strong potential for incorporation into more advanced merging techniques.

## D.3 EXPERIMENTS ON TOKEN-CHOICE VS EXPERT-CHOICE ROUTING

We also demonstrate our merging approach with the following routing mechanisms. We compare the baseline performance (i.e., the ties merging expert without Curvature Aware) under different routing mechanisms with the corresponding Curvature-Aware counterparts to see how different routing functions affect CAMEx performance. It is worth noting that Expert Choice routing is not compatible with the experts merging method, as discussed by Lory (Zhong et al., 2024) in their Subsection 5.3.:

- Stable MoE routing (Dai et al., 2022).
- Naive routing (Shazeer et al., 2017).
- X-MoE routing (Chi et al., 2022).

Note that the Curvature Aware model leverages the segment routing strategy (the causal segmenting strategy) proposed in Lory (Zhong et al., 2024), enabling a direct comparison between our model and the expert choice method. The results in Table 9 suggest that Curvature-Aware merging benefit from more advanced routing strategies. The CA model consistently outperforms the baseline with Ties merging across all routing mechanisms. Additionally, we observe that both Naive routing CA and X-MoE routing CA deliver robust performance across GLUE tasks, while Stable MoE routing CA emerges as the most reliable choice overall. In Table 8, the baseline models, including Vanilla, SMoE, and non-CA versions of Ties and Dare, achieve solid results but show diminishing improvements as model complexity increases. In contrast, our curvature-aware methods significantly outperform their counterparts. For instance, on the SQuAD dataset, Dare-CA achieves the highest Exact Match (EM) score of 81.76% and an F1 score of 88.60%, surpassing all other methods. Similarly, on WikiQA, Ties-CA attains the highest accuracy of 96.55%, with Dare-CA closely following at 96.23%.

Table 8: Performance of T5-base variants on question answering tasks.

| Methods | Params | TFLOPs | SQuAD Em/F1 | WikiQA Accuracy |
|---|---|---|---|---|
| Vanilla | 222M | 2.86 | 81.01/88.14 | 96.06 |
| SMoE | 1.0B | 2.86 | 81.25/88.50 | 96.04 |
| Domain-specific | 1.0B | 2.86 | 80.21/87.44 | 95.32 |
| Ties | 1.0B | 2.86 | 80.76/88.11 | 95.87 |
| Dare | 1.0B | 2.86 | 80.88/88.03 | 96.01 |
| **Domain-specific-CA** | 1.0B | 2.86 | 80.44/87.69 | 95.72 |
| **Ties-CA** | 1.0B | 2.86 | 81.52/**88.60** | **96.55** |
| **Dare-CA** | 1.0B | 2.86 | **81.76/88.60** | 96.23 |

Table 9: Performance of T5-base variants on the finetuning GLUE tasks

| Method | MRPC | RTE | STSB | SST-2 |
|---|---|---|---|---|
| Expert Choice MoE | 93.10 | 66.78 | 89.19 | 93.80 |
| Stable MoE routing Ties | 91.92 | 75.48 | 89.48 | 93.37 |
| **Stable MoE routing CA** | 92.96 | 78.76 | 89.64 | 94.63 |
| Naive routing Ties | 91.44 | 75.54 | 88.58 | 93.92 |
| **Naive routing CA** | 92.49 | 78.70 | 89.56 | 94.61 |
| X-MoE routing Ties | 91.99 | 75.29 | 88.42 | 93.26 |
| **X-MoE routing CA** | 92.79 | 78.20 | 89.26 | 94.38 |

## D.4    LONGER TRAINING FOR WIKITEXT-103 PRE-TRAINING

We conduct additional experiments by training for longer iterations on the Wikitext-103 dataset. The performance gaps between methods remain stable starting around epoch 40.

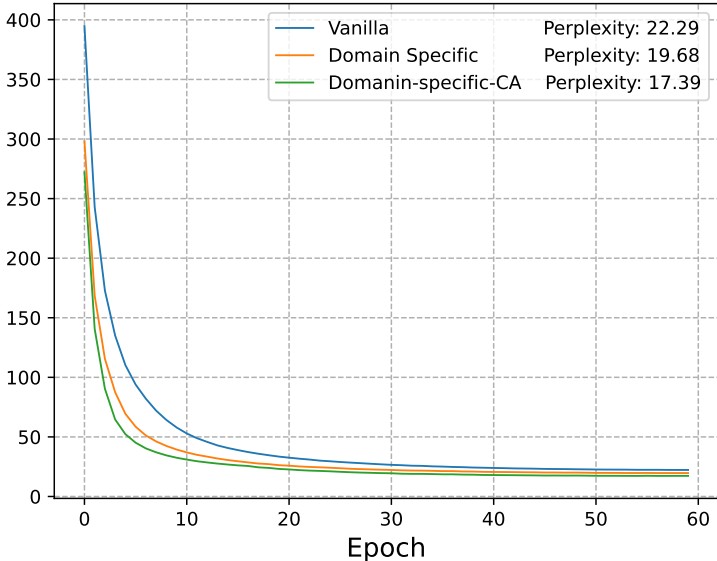

Figure 6: Performance of Vannila, Domain-specific, Domain-specific-CA under longer pre-training.

As shown in Figure 6 the trends demonstrate consistent improvements of our method over the baseline, with the gap remaining significant even after prolonged training.

## D.5 MORE COMPREHENSIVE ABLATION STUDY ON HYPERPARAMETERS

### D.5.1 ABLATION STUDY ON $\alpha$

We extend the range of $\alpha$ for the ablation study, specifically evaluating Dare-CA and Ties-CA with $\alpha \in [0.1, 1.6]$. The evaluation is conducted using 5 different seeds, and the results are averaged.

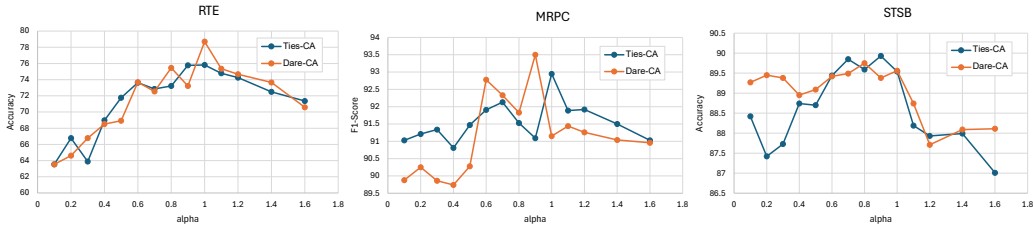

Figure 7: Test performance of Curvature-Aware methods under varying settings of $\alpha$.

The results in Figure 7 lead to the following observations:

- The performance of the models is suboptimal or even worse than the vanilla baseline when $\alpha$ is either too small ($\alpha \in [0.1, 0.4]$) or too large ($\alpha > 1.1$).

- Dare-CA is more sensitive to the choice of $\alpha$, showing sharper improvements and declines across the range.

- Ties-CA exhibits more gradual changes, suggesting it is more robust to variations in $\alpha$. The optimal range for $\alpha$ is $[0.8, 1.0]$.

### D.5.2 ABLATION STUDY ON NUMBER OF EXPERTS

We conduct additional studies on our method using different numbers of experts in the T5 backbone.

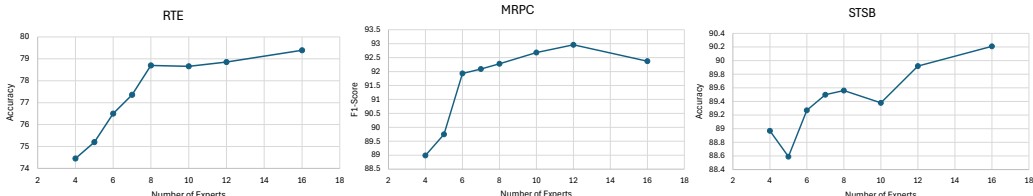

Figure 8: Test performance of Curvature-Aware methods under varying number of experts.

The following conclusions can be drawn from Figures 8:

- Increasing the number of experts generally improves accuracy up to a certain point: Accuracy improves as the number of experts increases, with the most significant gains occurring from 4 to 8 experts.

- After 12 experts, the accuracy either saturates or slightly decreases.

- We suggest using eight experts, as it provides a balanced trade-off between performance and efficiency.

## D.6 RESULTS ON LARGE BACKBONE (PHI-3)

To evaluate the effectiveness of CAMEx with large backbone, we experiment with Phi-3.

Table 10: Performance of Phi-3-mini variants on the fine-tuning tasks for the GLUE benchmark.

| Model | Params | SST-2 | MRPC | CoLA | STSB | RTE | QQP | QNLI | MNLI |
|---|---|---|---|---|---|---|---|---|---|
| Phi-3 | 3.8B | 95.76 | 90.12 | 61.36 | 88.7 | 80.51 | 92.38 | 94.84 | 90.39 |
| Phi3-Ties | 7.4B | 96.56 | 92.25 | 62.33 | 89.99 | 87.73 | 94.13 | 95.58 | 91.28 |
| Phi3-Ties-CA | 7.4B | **97.23** | **94.04** | **63.43** | **90.27** | **88.09** | **94.80** | **95.82** | **92.13** |

- The Ties-CA and Ties variants remarkably outperform the vanilla version, creating a substantial performance gap.
- Ties-CA further enhances the performance of Ties in all listed tasks.

Thus, we believe that curvature awareness has potential for improving other language models (LMs).

