# OpenReview forum: "CAMEx: Curvature-aware Merging of Experts"
_ICLR.cc/2025/Workshop/MCDC — MCDC @ ICLR 2025_

### Official Review · Reviewer_ud9b · 2025-02-26

**Rating:** 6
**Confidence:** 3
**Fit:** 5

**Summary:**

The paper proposes CAMEx, a method to for merging experts in (sparse) mixture-of-expert models. CAMEx improves over prior work by taking an approximation of the curvature of the parameter space into account.

Additionally, the authors propose a "dynamic merging" strategy which reduces the number of parameters while keeping FLOPs the same by merging into a global expert shared expert shared across layers.

CAMEx and the dynamic merging strategy are evaluated via a broad range of experiments on natural language as well as image tasks, and shown to improve over prior merging methods.

**Reason For Giving A Higher Score:**

The paper introduces & extensively evaluates a novel method which could be of interest to the community.

**Reason For Giving A Lower Score:**

-

**Strengths And Weaknesses:**

Strengths:
- The proposed method achieves consistent improvements on a broad set of tasks.
- Experiments are extensive & reported in a detailed way.

Weaknesses:
- The paper is lacking in clarity. The experimental setup is not clearly explained (i.e., what "Vanilla", "SMoE", and "Domain-Specific" refers to in Tables 3 and 4) and the dynamic merging strategy is only introduced in a Figure and via a formula but never explained in text.
- The paper skips comparing against some prior methods taking second order information into account, e.g. [[1]]. The authors' reasoning is that "[these approaches] become costly for large models, as storage and transmission demands increase linearly with model size and the number of tasks as well as the number of experts. Therefore, we choose baselines that are needless of extra information and computational cost to perform comparisons.". However, the proposed method also incurs extra computational cost & needs extra information so this seems like a weak argument.

[1]: https://arxiv.org/abs/2111.09832

**Suggestions:**

- Please introduce the experimental setup more clearly. As mentioned above, currently, what precisely the different baselines refer to is hard to understand.
- The dynamic merging method should also be given some more space to explain it more clearly.
- I would also recommend moving some of the content at the start of the introduction to related work, since it is essentially an enumeration of related work (Lines 33 - 43). This could make the introduction easier to read & more to the point.

---

### Official Review · Reviewer_7ENf · 2025-02-27

**Rating:** 7
**Confidence:** 3
**Fit:** 5

**Summary:**

The paper introduces CAMEx, a novel technique for merging experts in Sparse Mixture of Experts (SMoE) architectures that leverages natural gradients to incorporate the curvature of the parameter manifold. By moving beyond traditional Euclidean-based merging methods, CAMEx proposes a dynamic merging architecture that not only improves performance—demonstrated through experiments on different tasks such as language modeling (WikiText), text classification (GLUE), and image classification (ImageNet)—but also achieves computational efficiency with reduced memory overhead. The method is supported by both theoretical insights and empirical validations, positioning it as a promising direction for scalable and efficient model training.

**Reason For Giving A Higher Score:**

I give a higher score because the paper presents a novel curvature-aware merging method that leverages natural gradients to improve model performance and efficiency. Its strong theoretical foundation and empirical results across diverse tasks underscore its potential impact in scalable model training.

**Reason For Giving A Lower Score:**

I give a lower score because key experimental evidence is hidden in the appendix, reducing clarity and impact. Additionally, inconsistent model choices and a lack of statistical validation weaken the robustness of the claims. These issues suggest that further refinement is needed before the approach can be broadly adopted.

**Strengths And Weaknesses:**

**Strengths**

The paper's primary strengths lie in its novel use of curvature-aware natural gradients for expert merging, which leads to improved performance on different tasks/domains compared to traditional Euclidean methods.The methods is theoretically well-explained and empirically tested across different domains and tasks.

**Weaknesses**

The paper relegates many key experimental results and supporting figures to the appendix, which diminishes the clarity and impact of its main claims. There is an inconsistency in the use of models—switching between T5 and GPT-2 across different evaluations—without a clear justification, thereby undermining the demonstration of generalizability. Additionally, the lack of significance testing for the improvements in the curvature-aware variants weakens the confidence in the reported gains. Finally, the experiments are primarily focused on general datasets, with limited evaluation on domain-specific tasks, and the image classification results, a claimed contribution, are not prominently featured in the main text.

**Suggestions:**

- Main Text Integration: Move key figures, tables, and experimental results that support the paper's primary claims from the appendix to the main text for greater clarity and impact.

- Consistency in Model Choice: Clarify why the experiments switch between T5 and GPT-2 (e.g., GLUE versus Wikitext perplexity) and consider using both models consistently on the same tasks to better demonstrate generalizability.

- Statistical Significance: Include significance testing for the improvements of the curvature-aware variants (as shown in Tables 2, 3, and 4) to robustly validate the novel approach.

- Domain-Specific Evaluation: Extend experiments to more domain-specific datasets—such as those in math, coding, finance, healthcare, or non-English languages (e.g., Korean, Arabic)—to assess performance across varied contexts.

- Image Classification Results: Since image classification is claimed as a key contribution, integrate these results into the main text rather than keeping them exclusively in the appendix.

- Convergence Claims: Ensure that claims regarding rapid convergence are substantiated by results presented in the main text

- Provide clear motivations for the choice of models and datasets.

**Minor Comments**

- Table 1: Rescalling factor -> Rescaling factor
- Table 2: Use the same order for -CA counterparts: Swap the rows for DARE-CA and Ties-CA to be in the same order as the non-CA ones.

---

### Official Review · Reviewer_LSvg · 2025-02-28

**Rating:** 6
**Confidence:** 5
**Fit:** 4

**Summary:**

The paper proposes a curvature-aware merging method and the experiments show that CAMEx can outperform Euclidean-based expert merging. Overall, it is a very interesting idea, I suggest the authors to conduct more experiments on "combination of the skills". For example, one expert trained from the math dataset, another expert trained from the code dataset, it will be interesting if the curvature-aware merging can help to solve the math problems using code.

**Reason For Giving A Higher Score:**

I think the method is very interesting for expert or model merging. It would be good to conduct more experiments in this direction.

**Reason For Giving A Lower Score:**

no

**Strengths And Weaknesses:**

# Strengths

- The idea is interesting and the paper is presented clearly.
- The experimental results are promising to solve complex tasks.

# Weaknesses

- The evaluation benchmark is limited. It would be good to conduct more analysis on zero-shot tasks.
- Is the improvement of the results from the "Curvature-Aware" information? It would be good to conduct more analysis results.
- The results and conclusions are only based on T5. Although there are also some results based on Phi-3, it would be good to have more results based on other backbone models.

**Suggestions:**

As I said from the weakness.

---

### Decision · Program_Chairs · 2025-03-06

**Decision:**

Accept

**Comment:**

This work proposes a new merging method, which is very relevant to this workshop. All reviewers recommend acceptance, and we're pleased to accept it to this workshop.